# Mathematical determination of the HIV-1 matrix shell structure and its impact on the biology of HIV-1

**Weijie Sun**[1], **Eduardo Reyes-Serratos**[1], **David Barilla**[1], **Joy Ramielle L. Santos**[2], **Mattéa Bujold**[2], **Sean Graves**[3], **Marcelo Marcet-Palacios**[1,2]*

**1** Department of Medicine, Alberta Respiratory Centre, University of Alberta, Edmonton, Alberta, Canada, **2** Department of Biological Sciences Technology, Laboratory Research and Biotechnology, Northern Alberta Institute of Technology, Edmonton, Alberta, Canada, **3** Department of Mathematical and Statistical Sciences, University of Alberta, Edmonton, Alberta, Canada

* marcelo@ualberta.ca

**Data Availability Statement:** All relevant data are within the manuscript and its Supporting Information files.

## Abstract

Since its discovery in the early 1980s, there has been significant progress in understanding the biology of type 1 human immunodeficiency virus (HIV-1). Structural biologists have made tremendous contributions to this challenge, guiding the development of current therapeutic strategies. Despite our efforts, there are unresolved structural features of the virus and consequently, significant knowledge gaps in our understanding. The superstructure of the HIV-1 matrix (MA) shell has not been elucidated. Evidence by various high-resolution microscopy techniques support a model composed of MA trimers arranged in a hexameric configuration consisting of 6 MA trimers forming a hexagon. In this manuscript we review the mathematical limitations of this model and propose a new model consisting of a 6-lune hosohedra structure, which aligns with available structural evidence. We used geometric and rotational matrix computation methods to construct our model and predict a new mechanism for viral entry that explains the increase in particle size observed during CD4 receptor engagement and the most common HIV-1 ellipsoidal shapes observed in cryo-EM tomograms. A better understanding of the HIV-1 MA shell structure is a key step towards better models for viral assembly, maturation and entry. Our new model will facilitate efforts to improve understanding of the biology of HIV-1.

## Introduction

Over the last few decades, the scientific community has reported tremendous structural heterogeneity in type 1 human immunodeficiency virus (HIV-1), involving factors such as number of cores per particle, shape of the capsid cores and virion size. This structural diversity was documented by Johnson *et al.*, in their efforts to develop a structurally accurate mesoscale model of HIV-1 [1]. Given the unique nature of individual virions, development of a consensus model for HIV-1 is unlikely. However, as we increase our general understanding of structural irregularities, a more refined model will shed light upon unresolved mechanisms in viral particle formation, maturation and viral entry.

**Funding:** This work was supported by Alberta Innovates - Technology Futures grant G2015000484 to MMP. The funder had no role in study design, data collection and analysis, decision to publish, or preparation of the manuscript.

**Competing interests:** The authors have declared that no competing interests exist.

The most obvious variable of the HIV-1 virion is its size. The HIV-1 diameter ranges from 100 to 200 nm and as a consequence, the number of matrix (MA) and capsid (CA) proteins per virion also varies. On average, it has been estimated that mature HIV-1 particles contain 2000 to 5000 MA monomers [2]. The range in size can be affected by the preparation process for the virus, and by the number and condition of the capsid, ranging from ill-defined cores to virions containing 2 or more cores [3].

Though variable in size, the HIV-1 particle does follow specific structural patterns. Alfadhli *et al*. examined the quaternary structure of the HIV-1 MA lattice [4]. The authors determined the structure of the putative MA shell using myristoylated MA monomers and myristoylated MA bound to the Gag CA domain visualized using transmission electron microscopy. In both cases, the authors reported the formation of flat MA layers consisting of hexamers of MA tri-mers. The hexameric MA model influenced our understanding of other aspects of the HIV-1 structure. For example, the authors postulated that in this type of configuration, MA interac-tions with the cytoplasmic domains of the transmembrane glycoprotein gp41 would restrict mobility of the envelope glycoprotein complex (Env or gp41/gp120) to hexameric holes. In another study, the hexameric model was used to interpret Env incorporation during viral assembly [5]. The authors concluded that the cytoplasmic tail (CT) domain of gp41 could be recruited to the MA hexameric gaps to explain the observed dominant-negative inhibition results. More recently, Tomasini *et al*, developed a computational model to simulate the for-mation of the MA lattice structure, also assuming a hexameric configuration as an end-point for the Gag shell [6]. Thus, our current structural models of HIV-1 have a direct impact on how we explain scientific observations. Model-based, hypothesis-driven research has directly impacted advances in our understanding of the biology of HIV-1.

Careful analysis of the resulting MA trimer-derived hexameric configuration, however, reveals the geometric lack of feasibility of a hexamer-based polyhedron. The geometric limita-tions of this structure can be described by Euler's formula [7] in which the number of faces (*F*), vertices (*V*) and edges (*E*) of a polyhedron satisfy the equation $F+V-E = 2$. As a conse-quence, a polyhedron constructed solely of regular hexagons cannot exist. Furthermore, to visualize this geometric limitation one can imagine three regular hexagons connected with a common vertex on a flat plane. This shape will have no curvature because the interior angles of a hexagon are 120$^{\circ}$ and therefore three of them lined up flat will take up all 360$^{\circ}$ with no room for bending.

Given the limitations of the existing structural models for HIV, we developed a novel model for the HIV-1 MA shell and used this model to predict the number of MA protein units needed to assemble mature virions for the full range of observed HIV-1 particle diameters (100–200 nm). Our model also unveils a novel mechanism in which MA-imposed restrictions on Env mobility and location could assist the process of viral entry and perhaps viral particle assembly.

## Materials and methods

### Modeling a myristoylated HIV-1 MA trimer

The N-terminally myristoylated matrix (MA) determined by solution NMR (PDB ID: 2JMG) [8] was used to extract the structure of the N-terminal domain. A second structure (PDB ID: 1HIW) was used as the source of the quaternary structure of an MA trimer [9]. Thus, the myr-istoyl-group attached to amino acids GARS at the N-terminal of MA was combined with the amino acid sequence VLSG at the N-terminal end of 1HIW to generate an N-terminally myris-toylated MA trimer used in all MA models developed for this manuscript. All modeling

manipulations were conducted using python scripts run in PyMol (Molecular Graphics System, Version 1.7.6.0, Schrödinger, LLC.).

**Scripts and calculations.** All scripts and calculations were developed in the python language (Python Software Foundation. Python Language Reference, version 2.7 Available at http://www.python.org) and when necessary, coordinates were exported automatically to PyMol for construction of structures. PyMol natively uses Angstroms (Å) as its unit of length, thus all calculations were performed in Å. For the purposes of consistency and comparison with the structural biology literature, all units were reported in nanometers (nm) in this manuscript. All scripts are available for download at https://www.ualberta.ca/medicine/about/people/marcelo-marcet-palacios.

## Construction of lipid bilayers

A collada DAE file representing a sphere was created in Blender (Blender Foundation. Version 2.73. Available at http://www.blender.org). An Ico Sphere was created with subdivisions set to 4. The sphere size and its relation to the radius of a final lipid bilayer was determined empirically and shown to follow a linear curve (y = 0.198x – 4.728) with an $R^2$ = 0.998. The x-axis values used for Ico Sphere sizes were between 100 and 250 (arbitrary units) in Blender. The linear curve shown above might change outside of this x-axis range. The file was then imported to the software LipidWrapper [10] to generate the desired lipid bilayer structures. The sample red blood cell lipid bilayer was kindly provided by Shahinyan *et al.* [11] and available for download at www.bioinformatics.sci.am/Downloads/Erythrocyte.pdb.

## Mathematical algorithm used in the generation of coordinates for vectors to assemble the HIV-1 MA shell

A list *K* of "keeper" vectors was constructed where the center-of-mass of MA trimers would result. Vectors in *K* were found by rotating existing keeper vectors 120$^o$ and 240$^o$ around other "pivot" keeper vectors to obtain two potentially new keepers. It was then checked whether these new vectors were at least distance *r* (Formula 1 below) from all previously obtained keepers. The new keepers that met these criteria were then added to *K*.

More specifically, each vector in *K* was given an index and was subsequently used as a pivot to find additional vectors belonging in *K*. To do this, we constructed a second list *P* to indicate which vectors in *K* had already been used as pivots. While |*P*|<|*K*|, we chose the vector from *K* with the smallest index that was not yet in *P* and called this vector $v_i$. We then rotated $v_{\lfloor(i-1)/2\rfloor}$ 120˚ and 240˚ about $v_i$ to obtain vectors $v_{2i+1}$ and $v_{2i+2}$ respectively. It was then checked whether $\|v_{2i+1}-v_k\|>r$ for all $v_k$ in *K*. If so, then $v_{2i+1}$ was added to *K*. Similarly, if $\|v_{2i+2}-v_k\|>r$ for all $v_k$ in *K*, then $v_{2i+2}$ was added to *K*. Finally, $v_i$ was placed into *P* and the process was repeated until |*P*| = |*K*|. It should be noted that the first six vectors in *K* were found separately while the computer program used the generalized process above to find all other keepers.

## Results

### Construction of HIV-1 MA shell

It was recently demonstrated that HIV-1 MA trimers could form hexameric rings [4]. Alfadhli, *et al.* concluded that the distance between the center-of-mass of adjacent trimers was 9.02 nm. Because of the nature of the experiment, the generated symmetric patterns were created on a flat surface. We reconstructed this geometric arrangement (Fig 1A) to test the flat model predicted by Alfadhli, *et al.* [4], and we also reconstructed a cylindrical model predicted in the tubular arrays observed by Bharat *et al.* [12]. Using our computational methods, flat and

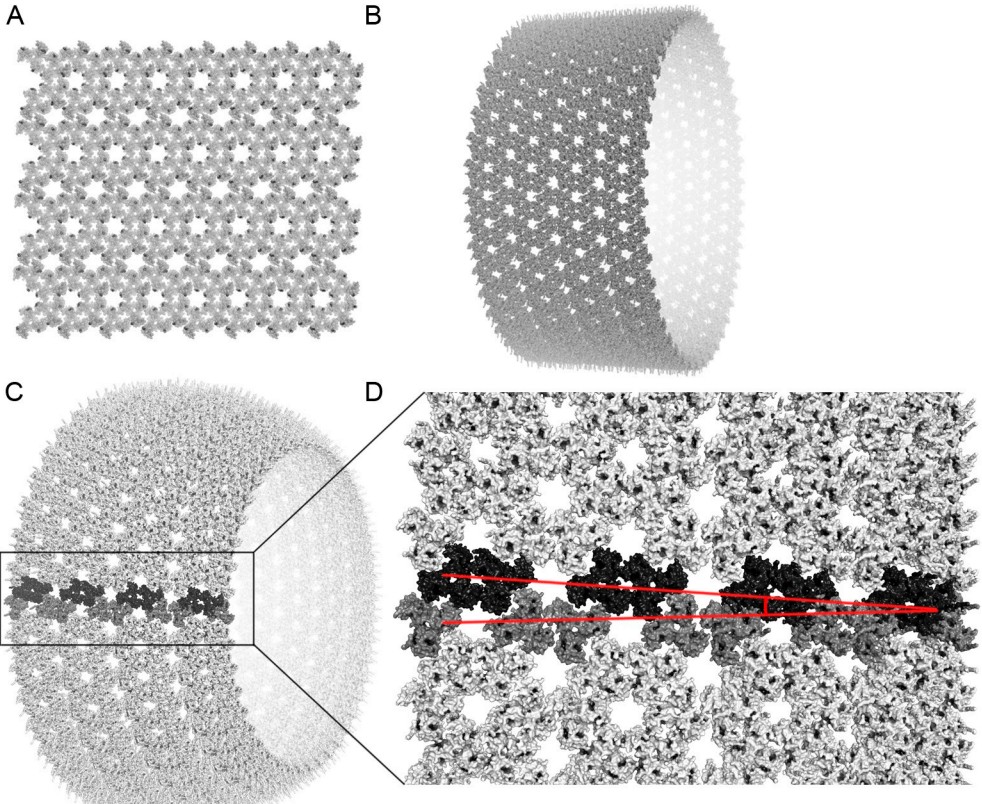

**Fig 1. HIV-1 matrix geometric configuration models.** (A) Flat hexameric configuration of the HIV-1 MA trimers. (B) Alternative cylindrical configuration of the HIV-1 MA trimer depicting geometric permissiveness of the configuration. (C) Spherical configuration of the HIV-1 MA trimer. MA trimers in black and dark grey are coloured to highlight increasing overlap of their volumes. (D) Zoomed in region from (C) showing increasing overlap between black and dark grey MA trimers. Angle is shown with red lines.

cylindrical configurations are indeed permissive (Fig 1A and 1B). However, generating a 3D model of the HIV-1 MA shell in its spherical conformation using hexameric rings was impossible (Fig 1C and 1D). Fig 1C and 1D show how adjacent trimers, coloured dark grey and black, collapse into each other's volume as the hexameric pattern progresses along the curve of a sphere. The distance between adjacent trimers decreases from 5.21 nm down to 5.02, 4.43, 3.49 nm (S2 Fig). The mathematical proof to demonstrate how it is impossible to construct a sphere with regular hexagons is shown in S3 Fig.

To solve the structure of the spherical HIV-1 MA shell, we developed a mathematical algorithm described in the Materials and Methods. We used a matrix system to determine the location of adjacent MA trimers on the surface of a sphere, making the assumption that the distance between adjacent trimers was 9.02 nm as previously reported [4]. The radius ($R$) of the sphere was equal to 61.5 nm or larger to accommodate an HIV-1 core of 117.52 nm. $R$ was measured from the center of the spheres to the center-of-mass of individual trimers. The furthest distance between 2 outermost atoms of individual HIV-1 cores was calculated using structures 3J3Q and 3J3Y reported earlier [13]. To create models of the HIV-1 MA shell, we used spheres of radius $R$ comprised of trimers whose center-of-mass points were at distance $r'$. Therefore, $r'$ was the shortest distance between adjacent trimers calculated using formula 1 (Fig 2A). This formula assumes that HIV-1 matrix trimers are arranged on a flat surface [4].

Mathematical derivation of this formula is included in S1A and S1B Fig.

$$r' = \sqrt{\frac{\left(\frac{9.02 \; nm}{2}\right)^2}{0.75}} = \frac{9.02 \; nm}{\sqrt{3}} \tag{1}$$

However, distance $r'$ is a function of the diameter of an HIV-1 particle, given that a matrix exists within a curved environment following a spherical geometry. Formula 2 calculates $r$, which is distance $r'$ adjusted to account for the spherical matrix (Fig 2B). Mathematical derivation of this formula is included in S1C and S1D Fig.

$$r = \sqrt{\left(\left(R - \sqrt{(R^2 - r'^2)}\right)^2 + r'2\right)} \tag{2}$$

The mathematical models for the HIV-1 MA shell were created using four conditions: (1) Adjacent MA trimers were separated by distance $r$ (e.g. 5.21 nm). (2) Empty spaces where the algorithm was unable to fit a trimer due to lack of available space were left vacant, even if a free-floating trimer or MA monomer could fit within the constraints of the gap. (3) All models were generated using a single MA trimer as a starting point or seed to start the algorithm. (4) For simplicity, all models were generated by rotating first 120° and then 240° around a pivot vector.

Using distances $r$ from a range of $R$ values, a rotational matrix was used using formula 3, where $\theta = 120°$ or $240°$ and $(u_x, u_y, u_z)$ is a normalized pivot vector. We have included additional information on the rotational matrix formula in S4 Fig.

$$R_m = \begin{bmatrix} \cos\theta + u_x^2(1 - \cos\theta) & u_x u_y(1 - \cos\theta) - u_z \sin\theta & u_x u_z(1 - \cos\theta) + u_y \sin\theta \\ u_y u_x(1 - \cos\theta) + u_z \sin\theta & \cos\theta + u_y^2(1 - \cos\theta) & u_y u_z(1 - \cos\theta) - u_x \sin\theta \\ u_z u_x(1 - \cos\theta) - u_y \sin\theta & u_z u_y(1 - \cos\theta) + u_x \sin & \cos\theta + u_z^2(1 - \cos\theta) \end{bmatrix} \tag{3}$$

The 3D coordinates obtained by multiplying keeper vectors by $R_m$ were plotted (Fig 3A). Starting with two original MA trimers on the surface of a sphere, additional trimers were included thus their relative connections and relative angles calculated and plotted (Fig 3B). These coordinates were then used to reconstruct HIV-1 MA shells. All models generated consist of six-spherical lune structures forming a 6-sided hosohedra (Fig 3C and 3D). Each lune was held in place by a central MA line (Fig 3E and 3F, dark gray trimers) that ran along the center of lune structures. Under all conditions tested for sphere diameters (range 22–222 nm, at 0.02 nm intervals for a total of 10,000 possibilities) and $r$ values (e.g. 5.21 nm ±2%, at 0.1% intervals for a total of 40 possibilities), there was no combination within the 400,000 spheres that resulted in adjacent lune structures that could connect with each other along their separating half great circles. Therefore, all generated spheres had the same fully separated 6-lune hosohedra configuration disconnected from adjacent lunes. For the purpose of this manuscript, we focused on particles with $r$ values of 5.21 nm. The raw data for these 10, 093 particles is provided in S1 File.

## Geometric considerations of 6-lune hosohedra structures in the viral entry process

In the structure described above, all 6 lunes connected at a single MA trimer at the origin (north pole) of the sphere. A resulting property of this shape is that during bud formation or viral entry, the shape of the sphere can change considerably during flat-to-spherical transitions and vice versa. This is indeed a property observed for viral entry and geometrically necessary

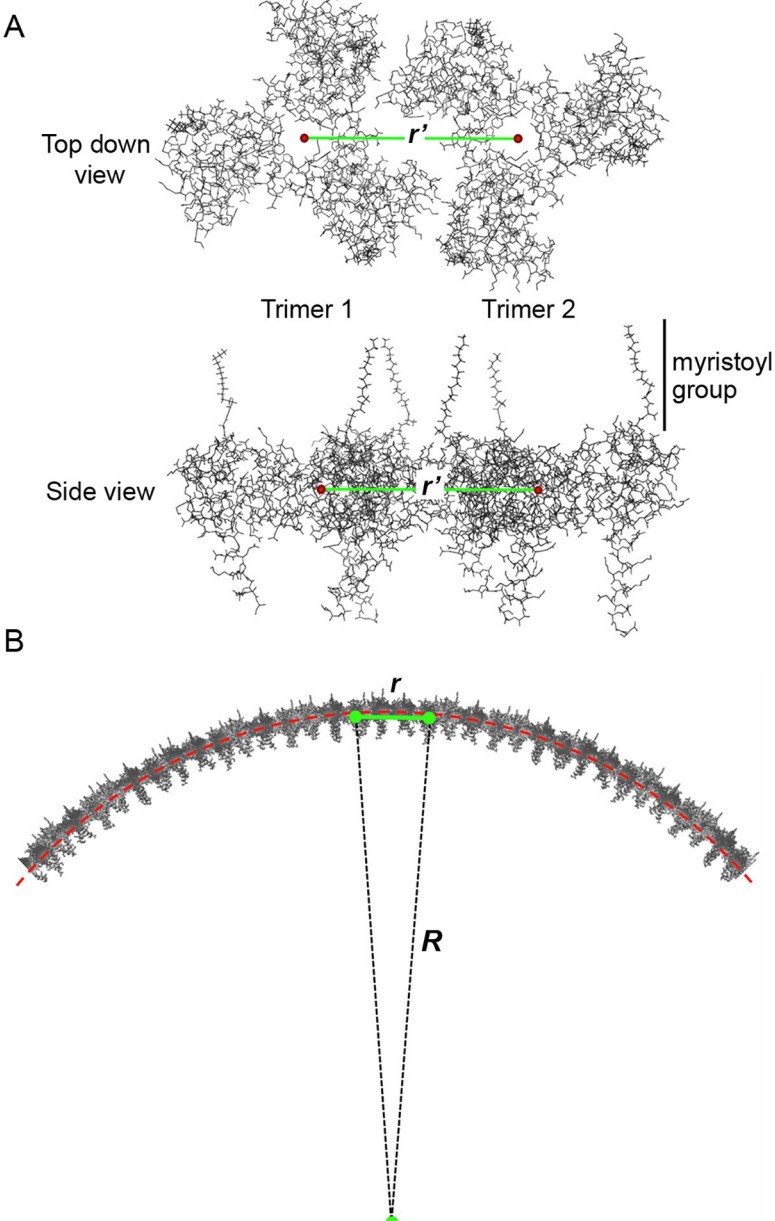

**Fig 2. Distance difference between adjacent MA trimers on flat and spherical planes.** (A) Representation of two adjacent MA trimers on a flat plane separated by distance $r'$. MA center-of-mass points (red) indicate the center-of-mass location for each trimer using a top-down view (top image) and side view (lower image). Myristoyl groups are more obvious on the lower image and labeled with a vertical line. (B) When MA trimers are embedded into the surface of a sphere (red dotted line), the distance between the center-of-mass of adjacent trimers changes (e.g. i.e. $r' \neq r$). In this panel we represent the radius of an MA shell sphere with $R$ and the distance between adjacent MA trimers with $r$.

to accommodate the Gag-polyprotein during bud formation. A second important characteristic of individual lune structures is that any central MA line generated side branches that never connected to another branch. Samples from this dataset are shown in Fig 4 at diameters 100, 150 and 200 nm. The resulting geometric arrangement confers the HIV-1 particles great geometric flexibility allowing for spherical-to-flat transitions of the MA shell (Fig 5). The model

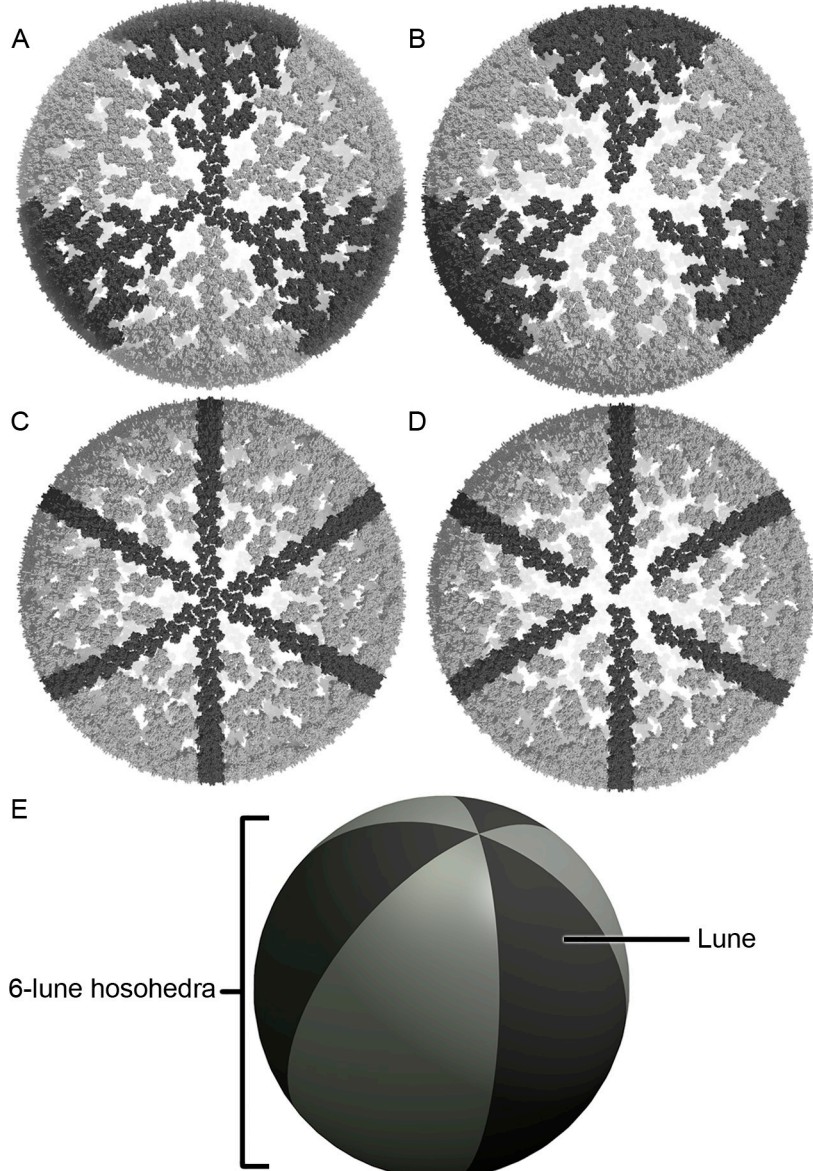

**Fig 3. Spherical arrangement of MA trimer proteins.** (A) Top-down view of the MA model centered at the starting MA trimer. Connected trimers are highlighted in dark grey or light grey to show a 6-lune configuration. (B) Bottom-up view of (A) showing how the 6 lunes do not reconnect. (C) Top-down and (D) bottom-up views of the MA sphere showing in dark grey the central MA line structure. (E) Spherical cartoon depicting a 6-lune hosohedra.

represented in Fig 5B illustrates how a 6-lune hosohedra MA configuration can transition from spherical (Fig 5A) to flat (Fig 5C), due to the disconnected nature of adjacent lunes.

## Relationship between HIV-1 particle diameter and number of MA polypeptides

To determine the validity of the 6-lune hosohedra MA model, we counted the number of MA individual units present in 400,000 HIV-1 MA spheres ranging in $R$-values from 22 nm to 222 nm. To these $R$ values, we added the distance from the center-of-mass of the MA trimers to

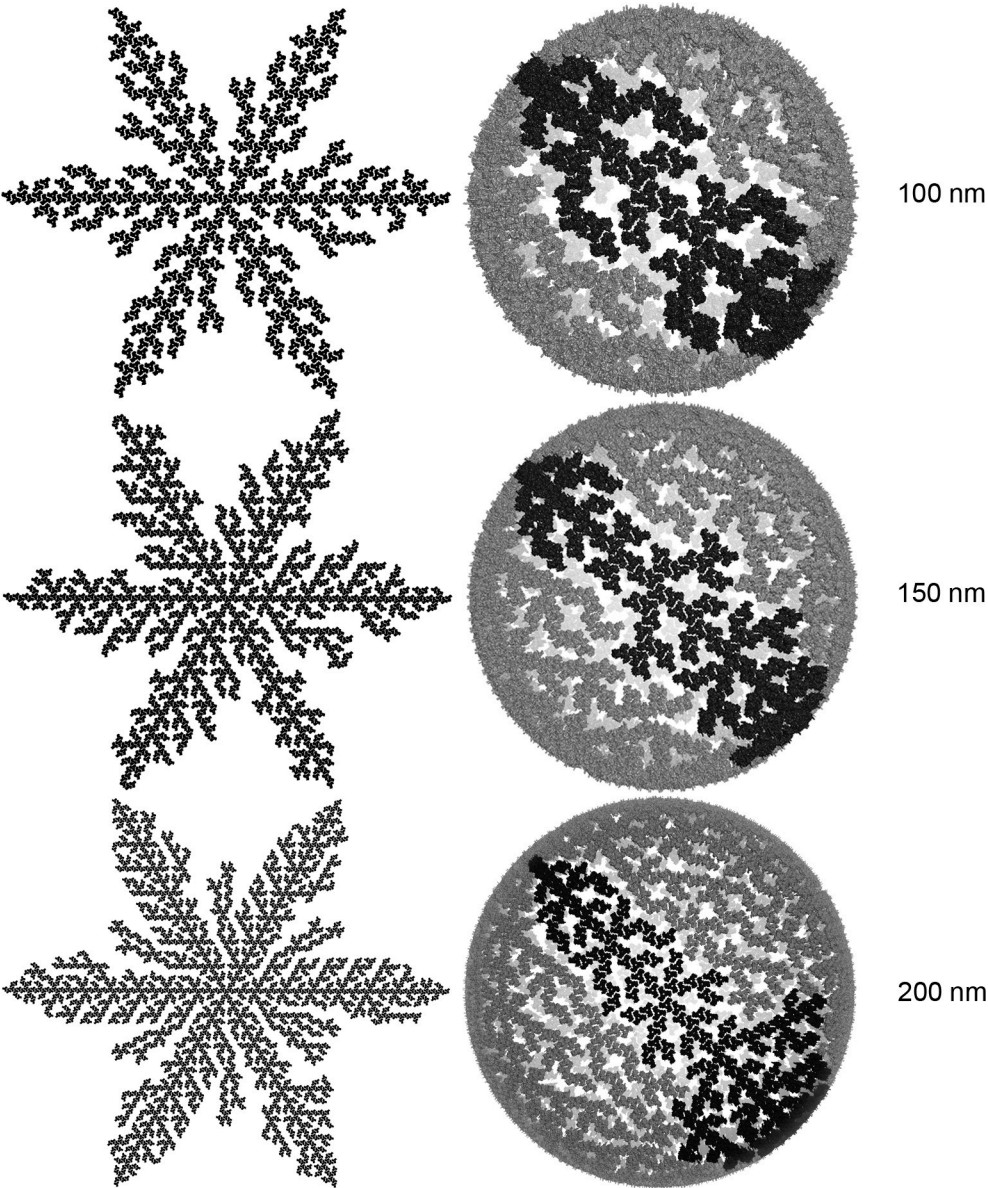

**Fig 4. Flat and spherical 3D arrangements of MA trimeric shells.** Structures were built at 100, 150 and 200 nm diameters. Flat lower-right lune structures were highlighted in dark grey in the corresponding sphere.

the point of contact with the lipid bilayer, (1.4 nm), and the width of the lipid bilayer (6 nm). Each of these values was then multiplied by 2 to account for the contribution of these distances on both sides of the sphere. Within this range, we plotted spheres with an HIV-1 diameter between 100 nm and 200 nm (Fig 6A). Reported HIV-1 particle diameter averages of 145 [14], 125.95 [15], 143 [2] and 125 [3] nm resulted in an overall average of 134.5 nm. We plotted the number of MA protein units for particles between 134 and 135 nm in diameter (Fig 6B). We observed that between 134.18 and 134.78 nm, the progressive increase of HIV-1 diameter did not result in any change on the total number of MA units, which remained constant at 2724 units per HIV-1 particle (Fig 6B).

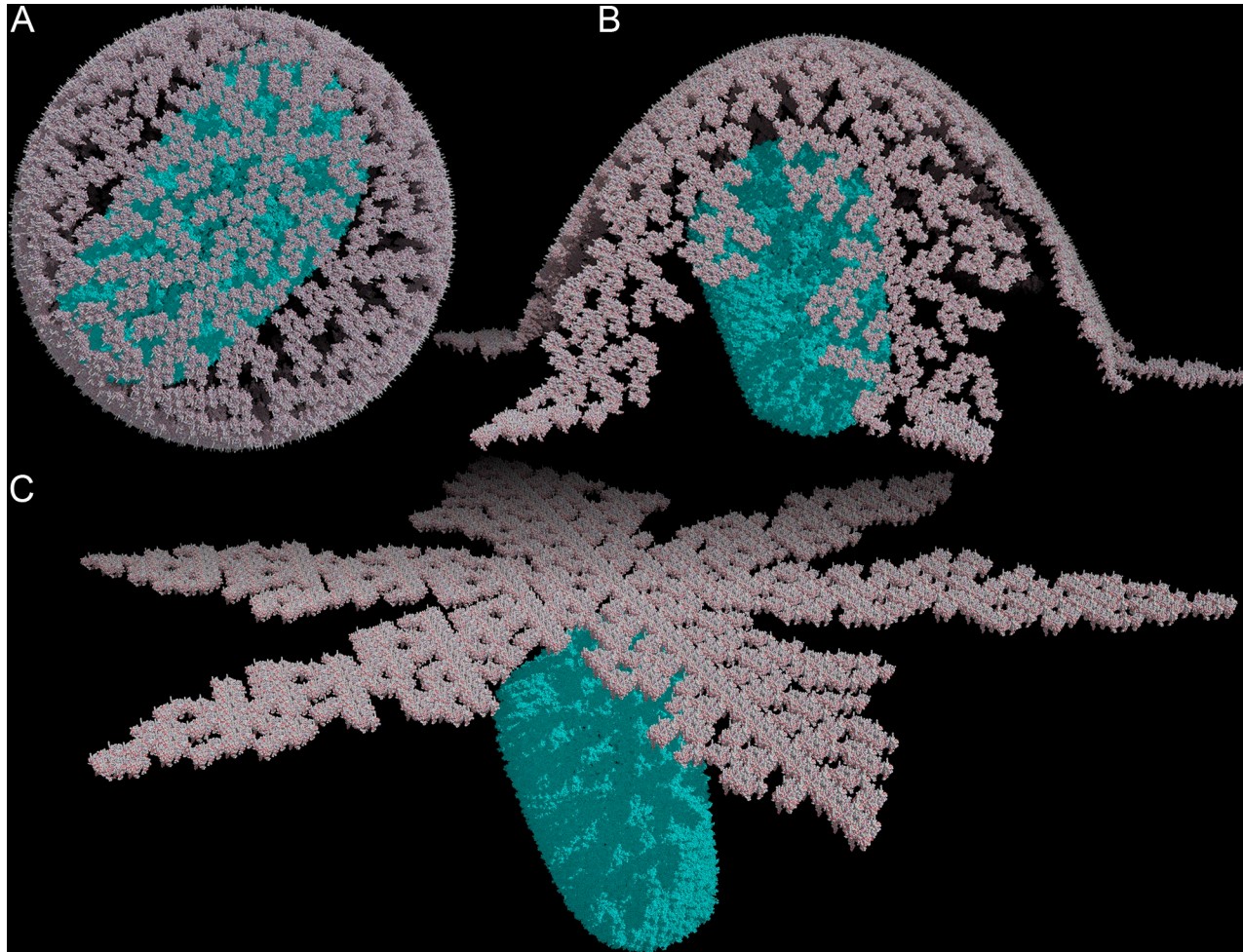

**Fig 5. Time-lapse of MA 3D structure during spherical-to-flat transition.** (A) HIV-1 MA in grey containing a viral core in blue (3J3Y). (B) Intermediate state during viral entry illustrating matrix lunes (grey) separation. (C) Final flat stage with fully extended MA shell. In this model, the host plasma membrane (not shown) would be positioned above the MA shell (grey) structure.

## Discussion

Due to the significant structural diversity in HIV-1 particles, it is unlikely that a single and accurate structural model of the virus will be fully explanatory. However, understanding the various structural characteristics of the pathogen is a powerful guide in our efforts to understand the pathophysiology of HIV-1. In this manuscript, we provide a mathematical explanation for how the HIV-1 MA shell cannot be arranged in a hexagonal configuration within a sphere. We also provide an alternative model that predicts an average 2724 MA particles in mature HIV-1 virions for HIV-1 particles of 134.5 nm. This number is much closer to the lower end of the estimated range of MA per virion of 2500, and not the higher end estimation of 5000 [6]. A third order (cubic) polynomial best-fit curve is provided to calculate MA unit numbers in HIV-1 particles of other sizes (Fig 6A). Interestingly, all MA shell models generated by our algorithm had a number of common features that would likely impact viral biology, most significantly, a 6-lune hosohedra configuration.

We make a case that HIV-1 particles are unlikely to be formed as hexagonal MA shells. It is important to recognize that addition of pentagons at perfectly symmetric intervals could result

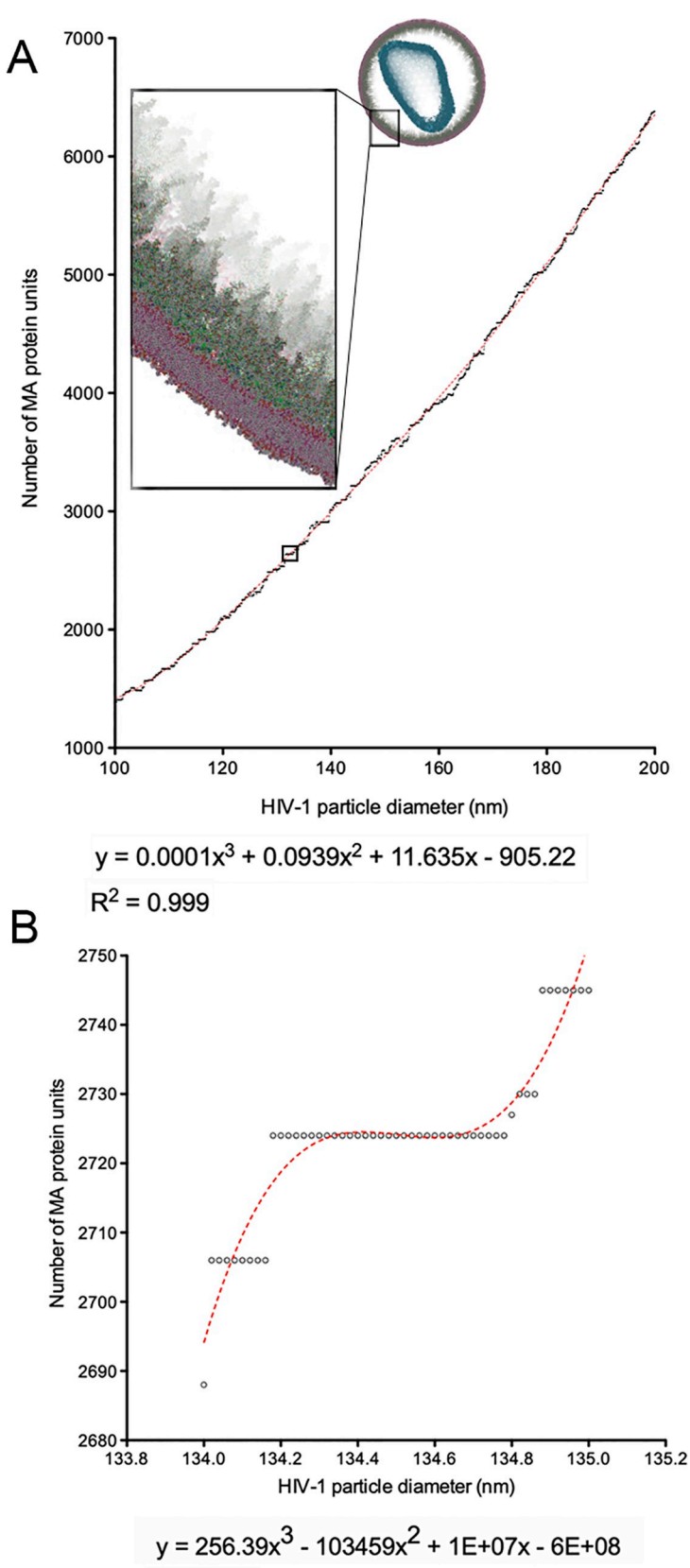

$$y = 0.0001x^3 + 0.0939x^2 + 11.635x - 905.22$$
$$R^2 = 0.999$$

$$y = 256.39x^3 - 103459x^2 + 1E+07x - 6E+08$$
$$R^2 = 0.896$$

**Fig 6. Relationship between the number of HIV-1 MA monomeric units and the diameter of HIV-1 particle.** (A) A total of 5000 HIV-1 particles consisting of a MA shell embedded into a lipid bilayer were generated for diameters between 100 and 200 nm. A third order (cubic) polynomial best-fit curve and its $R^2$ value are shown. The best-fit curve applies for diameters (x values) between 100 and 200 nm. Inset shows a representative HIV-1 particle with the core coloured blue, MA shell represented in green and the lipid bilayer represented in red/purple. Reported diameters include the lipid bilayers. A small region in the curve is highlighted with a square to indicate the portion of panel (A) expanded in panel (B). (B) 50 HIV-1 particles from (A) with diameters 134 to 135 nm. A third order (cubic) polynomial best-fit curve and its $R^2$ value are shown. The best-fit curve applies for diameters (x values) between 134 and 135 nm.

in hexagon-based spheres (e.g. patterns of a traditional soccer ball or fullerenes). However, this arrangement is unlikely to occur in the HIV-1 MA shell unless a molecular mechanism capable of strategically positioning pentagons existed. Failure to position pentagons in correct positions would result in non-spherical structures such as that of the HIV-1 capsid. HIV-1 capsids are built from ~216 hexamer and 12 pentamer units of CA [16]. Introduction of pentamer units into the geometry of the HIV-1 capsid allows for the necessary curvature required to enclose the core [17]. In their manuscript Liu *et. al.* demonstrated the complex interplay of Euler's theorem and the contribution of CA pentamers in the sharp declinations on various types of HIV-1 capsids. This strategy results in the well-known conical core structures with facets (hexagonal straight segments) joined by curved regions (pentagon-derived curves) [3]. The shape of the HIV-1 capsid illustrates what happens when pentamers are positioned in a non-repeating pattern resulting in non-spherical shapes.

Also, a perfectly continuous MA shell without gaps would lead to a rigid structure that is unlikely to accommodate more common ellipsoidal shapes observed in cryo-EM tomograms [18]. By contrast, the flexibility conferred by our model can accommodate for particle shapes that depart from perfect spheres and can also account for the significant increase in viral particle volume that takes place upon CD4 receptor engagement [18]. The reported 26% to 33% volume increase can be visualized in the sphere-to-flat shape transition (Fig 5A and 5B).

If the hexagonal MA shell structure of HIV-1 cannot exist in spherical configuration unless additional pentagons or spacers are precisely inserted into the structure, then the currently accepted model for HIV-1 particle assembly also needs to be revisited. The Gag polyprotein layer starting as a flat hexagonal arrangement cannot be geometrically bent into the curve of a sphere. Other models have shown that Gag-derived units are capable of forming tubular arrays in cylindrical configurations [12], which we confirmed to be mathematically permissive (Fig 1B). However, those cylindrical intermediary structures also cannot be geometrically converted into a sphere.

There are currently at least two methods developed to predict the mechanism of viral capsid assembly and packaging, namely the differential equation model and the coarse-grained modeling approach. The differential equation method was developed by Zlotnick *et. al.* [19, 20] and Hagan *et. al.* [21–23]. More recently, both methods have been used to model HIV-1 Gag during viral assembly. Using the differential equation method, Liu *et. al* [24] predicted a nucleating event containing a 7-mer and then a 13-mer (7 and 13 connected hexagons) initiating formation of the Gag sphere [24]. Similarly, using the coarse-grained method, Tomasini *et. al.* [6] predicted capsid assembly dynamics allowing hexameric configurations for the Gag shell. By contrast, our model predicts a new assembly method where the curve of the sphere emerges as new Gag trimers are added, starting from a single initial trimer. From the apex of that single trimer, our model predicts addition of trimers following the 6-lune structure, to create the growing curvature of the forming sphere. As the curvature of the HIV-1 particle is forming, new Gag trimers would attach to the forming shell in locations allowed by the existing steric hindrance of the structure. As the sphere volume closes onto itself following the

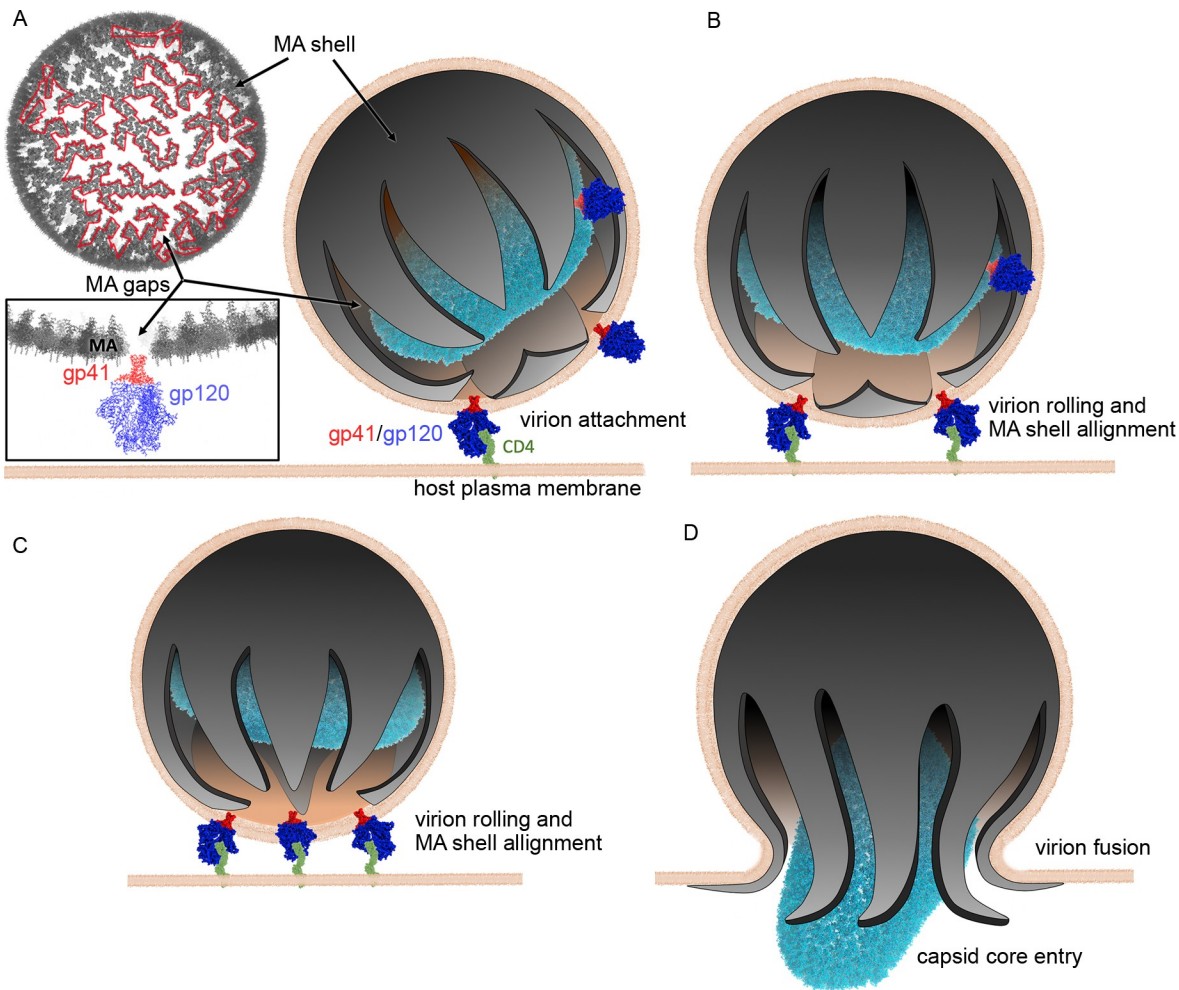

**Fig 7. Conceptual model of HIV-1 viral attachment and entry.** (A) Interconnected MA gaps (White area surrounded by a red border) are regions where Env heterotrimers (red gp41 and blue gp120) localize. MA gaps converge at the bottom of the structure creating a 6-lune MA shell. (B) HIV-1 particles bound to CD4 receptors (green) can spin and roll, thus changing the orientation of the virion particle in such a way as to align converging MA gaps with the host plasma membrane. (C) Binding of additional CD4 receptors increases virion alignment. (D) Flexible lunes facilitate viral capsid (light blue core) entry into the host.

6-lune pattern, the complete HIV-1 particle is formed (Fig 3). In agreement with our model, Briggs *et. al.* also observed that as curved Gag hexameric lattices grow, hexamers quickly become tightly packed [25]. In the model proposed by Briggs *et. al.*, regions of empty spaces (no Gag proteins present), significant variation in distances between adjacent hexagons and overlap of hexameric unit cells were needed to generate curved surfaces. These and other aspects of HIV-1 assembly and maturation have been reviewed earlier [26].

Our new model also has implications for understanding HIV-1 attachment and entry. We know that gp41/gp120 (Env) is important for viral cytopathicity and infection. Checkley *et al.* published a review on the roles of gp41 [27] including roles of the unusually large cytoplasmic tail (CT) domain. A demonstrated function of the CT domain is to aid in the incorporation of gp41 into the virion via interactions with the MA shell. Truncations in the CT region abrogate interactions of gp41 with MA [28, 29]. During particle formation, CT domains can induce clustering and reduce mobility of gp41 [30, 31]. The model we propose for viral entry predicts the MA shell to exist in a 6-lune structure, with MA gaps between lunes (Fig 7). We

hypothesize that the 3D architecture of the MA shell restricts the accessibility of gp41 to MA gaps, consistent with the clustering effect of CT domains on gp41 as shown in previous studies [30, 31]. Any gp41/gp120 within a MA gap can bind to a CD4 receptor on the host membrane. The attached virion can roll across the surface of the plasma membrane until more clustered gp41/gp120 bind to additional CD4 receptors, thus facilitating the alignment of the virion's interconnected MA gaps towards the host membrane (Fig 7). The intrinsic flexibility of the lunes themselves would promote virion fusion and capsid entry (Fig 7)

It should be pointed out that our model is subject to some limitations. It begins with MA trimers in isolation, without interactions with other proteins such as Env proteins. In fact, we know that Env proteins are incorporated with assistance of Gag units in the inner side of the plasma membrane. It is likely that Gag-Env complexes will have an effect on the final overall geometry of the MA shell, which is not included in our prediction model. Our model also assumes that all originally present MA trimers remain attached to the viral membrane past the maturation phase. Removal of MA units that could compromise the structural stability of a mature HIV-1 particle may be a necessary step in virus maturation. Currently, our model does not account for monomeric, dimeric, or trimeric forms of MA units disconnected from the main MA shell. Because our MA unit estimates (Fig 6) do not account for the presence of disconnected MA units, we anticipate that the values we report here might underestimate actual numbers. Despite these limitations, our model predicts a highly flexible shell, facilitating particle fusion and entry while still providing structural support. Our model provides a framework for future *in-silico* kinetic studies and empirical validation of the structure of the HIV-1 MA shell.

## Supporting information

**S1 Fig. Derivation of Formulas 1 and 2.** (A) The distance between the centers of mass of two adjacent trimers was obtained experimentally by Alfadhli *et al*. [4]. This distance is 9.02 nm and is represented by a red line. The construction process of our models required the distance between connected trimers (*r'* = trimers connected by black lines). (B) The resulting triangle assumes that trimers are arranged on a flat surface. This distance can be calculated using the formula shown. (C) However, trimers are arranged on the surface of a sphere, thus we move the center of mass of trimer from a plane to the surface of the sphere while maintaining distance *R*. to calculate distance *r*. (D) Equation to calculate *r* and its components is shown.
(TIF)

**S2 Fig. Impact of hexagonal arrangement of MA trimers within a sphere.** Starting from two adjacent trimers with green centers-of-mass separated by *r* = 5.21 nm (top trimers), we proceed with construction of the model following the direction of the green arrows to place trimers 3 and 4. Following the same procedure, trimers 5 and 6 are placed to complete the first hexagon (yellow centers-of-mass). This first hexagon is not a regular hexagon, as the final side (distance between yellow centers-of-mass) is reduced to 5.02 nm. If we proceed with creation of another adjacent hexagon, the last two trimers are separated by 4.43 nm, a distance that results in the collapse of these volumes into each other (blue centers-of-mass). Hence, it is impossible for 2 hexagons to co-exist side-by-side. The increasingly shorter ends of adjacent hexagons measure 5.02, 4.43, 3.49 nm and continues to decrease down from its original value of 5.21 nm.
(TIF)

**S3 Fig. Mathematical demonstration showing how regular hexagons cannot form 3-dimentional spheres.**
(TIF)

**S4 Fig. Description and use of formula 3.**
(TIF)

**S1 File. Generation of HIV-1 particles ranging in diameter between 24.80 and 226.64 nm.**
These particles were created in the software PyMol which uses Angstroms for units of distance, thus the initial range of radius distances were between 50.0 Å and 1059.2 Å (column A). Note that this radius was calculated to the center-of-mass of HIV-1 trimers. This distance was simply multiplied by two to calculate diameter in column B and converted to nm by multiplying the latter values by 0.1 (column C). The lipid bilayer was estimated to be 6 nm in width. To account for both sides of the sphere, this distance was multiplied by two and thus 12 nm was added to calculate the diameter including lipid bilayer width (column D). To calculate the final diameter of HIV-1 spheres, we also added the distance between the HIV-1 MA trimer center-of-mass to the inner surface of the lipid bilayer in column E. The number of MA trimers was obtained using the algorithms described in Materials and Methods to generate the data in column F. The number of MA monomers per HIV-1 particle was calculated by multiplying the values in column F by three (Column G). The data shown in Fig 6A and 6B were plotted from these data set.
(XLSX)

## Acknowledgments

We thank Stewart Cook (Dean of the School of Applied Sciences and Technology from NAIT) and Trevor April (Department Head of the Department of Natural Sciences and Academic Studies from NAIT) for providing download support. We thank David Christiansen (Supervisor–SAST Tech Services/CADD Labs) for graphic station construction and technical support.

## Author Contributions

**Conceptualization:** Weijie Sun, Eduardo Reyes-Serratos, David Barilla, Joy Ramielle L. Santos, Sean Graves, Marcelo Marcet-Palacios.

**Data curation:** Weijie Sun, Eduardo Reyes-Serratos, David Barilla, Marcelo Marcet-Palacios.

**Formal analysis:** Weijie Sun, David Barilla, Sean Graves, Marcelo Marcet-Palacios.

**Funding acquisition:** Marcelo Marcet-Palacios.

**Investigation:** Marcelo Marcet-Palacios.

**Methodology:** Weijie Sun, Eduardo Reyes-Serratos, David Barilla, Sean Graves, Marcelo Marcet-Palacios.

**Project administration:** Mattéa Bujold, Marcelo Marcet-Palacios.

**Resources:** Eduardo Reyes-Serratos, Mattéa Bujold, Marcelo Marcet-Palacios.

**Software:** Weijie Sun, Marcelo Marcet-Palacios.

**Supervision:** Marcelo Marcet-Palacios.

**Validation:** Marcelo Marcet-Palacios.

**Visualization:** Weijie Sun, Marcelo Marcet-Palacios.

**Writing – original draft:** Sean Graves, Marcelo Marcet-Palacios.

**Writing – review & editing:** Weijie Sun, Eduardo Reyes-Serratos, David Barilla, Joy Ramielle L. Santos, Mattéa Bujold, Sean Graves, Marcelo Marcet-Palacios.

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
