## [Decision Letter · Decision Letter 0]

10 Sep 2019

PONE-D-19-19426

Mathematical Determination of the HIV-1 Matrix Shell Structure and its Impact on the Biology of HIV-1

PLOS ONE

Dear Dr. Marcet-Palacios,

Thank you for submitting your manuscript to PLOS ONE. After careful consideration, we feel that it has merit but does not fully meet PLOS ONE’s publication criteria as it currently stands. Therefore, we invite you to submit a revised version of the manuscript that addresses the points raised during the review process.

This manuscript was reviewed by two experts in the field, who raised several points that need to be addressed carefully before a revised version is considered for publication. All comments are self explanatory and require more detailed description of techniques or results.        

We would appreciate receiving your revised manuscript by Oct 25 2019 11:59PM. To enhance the reproducibility of your results, we recommend that if applicable you deposit your laboratory protocols in protocols.io, where a protocol can be assigned its own identifier (DOI) such that it can be cited independently in the future. For instructions see: http://journals.plos.org/plosone/s/submission-guidelines#loc-laboratory-protocols

We look forward to receiving your revised manuscript.

Kind regards,

Jamil S Saad, Ph.D

Academic Editor

PLOS ONE

Journal Requirements:

Reviewers' comments:

Reviewer's Responses to Questions

**Comments to the Author**

1. Is the manuscript technically sound, and do the data support the conclusions?

Reviewer #1: Yes

Reviewer #2: Yes

2. Has the statistical analysis been performed appropriately and rigorously? 

Reviewer #1: Yes

Reviewer #2: Yes

3. Have the authors made all data underlying the findings in their manuscript fully available?

Reviewer #1: Yes

Reviewer #2: No

4. Is the manuscript presented in an intelligible fashion and written in standard English?

Reviewer #1: Yes

Reviewer #2: No

5. Review Comments to the Author

Reviewer #1: (1) The "6-lune hosohedra" structure should be mentioned in the abstract.

(2) Formula 1 & 3 look mysterious, there should be more explanation about

Formula 3.

(3) For the 2 paragraphs following Formula 3 "Geometric consideration ..."

and "Relationship between HIV-1 ...", there should be some tables about

percentage or distribution of the data you obtained for the 400,000 HIV-1

particles or MA spheres.

(4) Figures 3,4,5 are somewhat useful but not enough for understanding the

6-lune hosohedra structure. Is it possible to have some figures for

illustration in mathematical or geometric aspect?

(5) In Discussion Paragraphs 2 & 5, it is stated hexagonal MA shells are

unlikely and a comparison to CA hexagonal structure is presented. For

this regard, a recent paper might provide some useful info:

Jiangguo Liu, Farrah Sadre-Marandi, Simon Tavener, Chaoping Chen, Curvature

concentrations on the HIV-1 capsid, Mol. Based Math. Biol., 3(2015),

pp.43-53

(6) For Discussion Paragraph 3, I am glad that the authors mentioned

potential gaps among MA units. This should allow some random factors in

the formulation of MA shells.

(7) For Discussion Paragraph 7, it is helpful that the authors have

discussed the limitation of their model.

Overall, this paper is well written but needs further elaboration on

derivation of their 6-lune hosohedra model.

Reviewer #2: In this manuscript authors used geometrical and rotational matrix computational methods to construct a model and predict a new mechanism for viral entry. This work explained the increase in particle size observed during CD4 receptor engagement and the most common HIV-1 ellipsoidal shapes observed in cryo-EM tomograms. The work is interesting and meaningful. However, my concerns are listed as follows.

COMMENT #1.

In abstract, “Evidence by various high-resolution microscopy techniques support a model composed of MA trimers arranged in a hexameric configuration.”

Firstly, the evidence is only one side, but some evidences may support a model composed of MA hexamers. Obviously, MA hexamers consist of two MA trimmers. I believe that it may be difficult to differentiate them. Therefore, I suggest that authors should describe these two views in the introduction.

Secondly, this sentence is isolate to its surrounding sentences, and it is redundant, I think.

COMMENT #2.

Only one reference is shown about the coarse-grained method. I suggest that authors should cite more important works about it because it is one of key methods about virus assembly.

COMMENT #3.

Authors only presented one method (the coarse-grained method) to review the methods on HIV capsid assembly. It is not comprehensive. As far as I know, at least two main methods, including the coarse-grained method and ordinary differential equation model, are developed to simulate the HIV capsid assembly. The ordinary differential equation model assumed that the assembly model equilibrates rapidly. It was developed by Zlotnick, et. al. (J. Mol. Biol. 241 (1994), Biochemetry(1999)), and Hagan et. al. (Biophys. J. 98 (2010), Adv. Chem. Phys. (2013), Rev. Phys. Chem. 66 (2015)) to simulate viral assembly. More recently, it was applied to study HIV-1 capsid assembly by Liu, et. al. (Bull Math Biol (2019).

COMMENT #4.

“a 3-dimensional (3D) object constructed solely of regular hexagons cannot exist.” The meaning of this sentence may be not right. I can form a 3D capsid with hexagons by adjusting the gaps between hexagons. These irregular gaps may accommodate the curvature necessary to form an approximately spherical protein shell (E.O. Freed , Springer (2013), E.O. Freed Nat. Rev. Microbiol. (2015) )

COMMENT #5.

The sphere size and its relation to the radius of a final lipid Bilayer: y = 0.198x – 4.728. To my understanding, if x=10, the sphere size y<0. Therefore, this formula may be improved or authors may add some explanations about it.

There are still two similar formulas needed to be improved in this paper.

COMMENT #6.

“generating a 3D model of the HIV-1 MA shell in its spherical conformation using hexameric rings was impossible (Fig. 1C and D).” To my understanding, this method does not generate a spherical conformation using hexamers. But I believe that a spherical conformation can be generated by hexamers by adjusting gaps between hexamers.

COMMENT #7.

Please check the reference “14. Cole IR (2015) Modelling CPV. 1-260.”

6. PLOS authors have the option to publish the peer review history of their article (what does this mean?). If published, this will include your full peer review and any attached files.

Reviewer #1: No

Reviewer #2: No

---

## [Author Response · Author response to Decision Letter 0]

4 Oct 2019

Dear Reviewers,

We are thankful for your careful analysis and insightful observations. We have followed your comments and have: 

1) Enhanced figures 3 and 5

2) Added 4 supplemental figures and 1 supplemental table.

3) Enhanced the text of the manuscript taking into account your comments

4) Added 9 new references

Kind regards,

Marcelo Marcet-Palacios

Answers to Reviewer 1’s Comments:

Reviewer #1: (1) The "6-lune hosohedra" structure should be mentioned in the abstract.

We agree and have modified the abstract to include this information. It now reads: “In this manuscript we review the mathematical limitations of this model and propose a new model consisting of a 6-lune hosohedra structure, which aligns with available structural evidence.

(2) Formula 1 & 3 look mysterious, there should be more explanation about

Formula 3.

We have developed a new Supplemental Figure (Supp. Fig. 1) to explain the derivation and application of formulas 1 and 2. We have added a new sentence at the end of the second paragraph in the Results section: “Mathematical derivation of this formula is included in Supp. Fig. 1A and B.” and a second sentence at the end of paragraph 3 of the same section: “Mathematical derivation of this formula is included in Supp. Fig. 1C and D”.

We identified a typo in formula 1 and have now fixed this typo (i.e. R) with the correct value (i.e. 9.02 nm).

To help the readers understand formula 3 we have added a new Supp. Fig. 4.

(3) For the 2 paragraphs following Formula 3 "Geometric consideration ..."

and "Relationship between HIV-1 ...", there should be some tables about

percentage or distribution of the data you obtained for the 400,000 HIV-1

particles or MA spheres.

This was an excellent observation by the reviewer. We have included the raw data as Supplemental Table 1. In depth description of the data is included in the legend of the table. In short, the table shows in columns A through E the stepwise calculation of HIV-1 particle diameter and the corresponding number of MA trimers (column F) and number of MA monomers per virion (column G). This data was used to generate Fig. 6.

(4) Figures 3,4,5 are somewhat useful but not enough for understanding the

6-lune hosohedra structure. Is it possible to have some figures for

illustration in mathematical or geometric aspect?

We added a new panel E to Fig. 3 to help the reader understand the geometry of a 6-lune hosohedra. We removed original panels A and B in this figure, which were less effective in showing mathematical and geometrical aspects of the structure. We enhanced Fig. 5 by showing more consistent colors for all 3 panels and made the background black. 

(5) In Discussion Paragraphs 2 & 5, it is stated hexagonal MA shells are

unlikely and a comparison to CA hexagonal structure is presented. For

this regard, a recent paper might provide some useful info:

Jiangguo Liu, Farrah Sadre-Marandi, Simon Tavener, Chaoping Chen, Curvature

concentrations on the HIV-1 capsid, Mol. Based Math. Biol., 3(2015),

pp.43-53

We thank the reviewer for kindly sharing with us the manuscript by Jiangguo Liu et. al. We updated the second paragraph of the discussion by adding the descriptor “”fullerenes” and 2 sentences referencing Liu’s work. “Introduction of pentamer units into the geometry of the HIV-1 capsid allows for the necessary curvature required to enclose the core [18]. In their manuscript, Liu et. al. demonstrated the complex interplay of Euler’s theorem and the contribution of CA pentamers in the sharp declinations on various types of HIV-1 capsids. ” 

(6) For Discussion Paragraph 3, I am glad that the authors mentioned

potential gaps among MA units. This should allow some random factors in

the formulation of MA shells.

Thank you for this kind feedback.

(7) For Discussion Paragraph 7, it is helpful that the authors have

discussed the limitation of their model.

We appreciate the encouraging feedback provided by the reviewer.

Overall, this paper is well written but needs further elaboration on

derivation of their 6-lune hosohedra model.

Thank you for this comment. Our revisions have further elaborated on the derivation of the 6-lune hosohedra model, to provide better explanation for the reader.

Answers to Reviewer 2 Comments:

COMMENT #1.

In abstract, “Evidence by various high-resolution microscopy techniques support a model composed of MA trimers arranged in a hexameric configuration.”

Firstly, the evidence is only one side, but some evidences may support a model composed of MA hexamers. Obviously, MA hexamers consist of two MA trimmers. I believe that it may be difficult to differentiate them. Therefore, I suggest that authors should describe these two views in the introduction.

Secondly, this sentence is isolate to its surrounding sentences, and it is redundant, I think.

We thank the reviewer for drawing our attention to this point. We have improved the sentence in the abstract by adding: “…configuration consisting of 6 MA trimers forming a hexagon”. We hope this clarifies that our manuscript is specifically describing the trimer-hexagon model and therefore we focus our efforts in that regard. A new supplemental figure (Supp. Fig. 2) to better show the readers the structure of an MA hexamer composed of 6 trimers and thus a total of 18 MA monomers was added to the manuscript. 

COMMENT #2.

Only one reference is shown about the coarse-grained method. I suggest that authors should cite more important works about it because it is one of key methods about virus assembly.

We agree with this comment and have addressed this under COMMENT #3.

COMMENT #3.

Authors only presented one method (the coarse-grained method) to review the methods on HIV capsid assembly. It is not comprehensive. As far as I know, at least two main methods, including the coarse-grained method and ordinary differential equation model, are developed to simulate the HIV capsid assembly. The ordinary differential equation model assumed that the assembly model equilibrates rapidly. It was developed by Zlotnick, et. al. (J. Mol. Biol. 241 (1994), Biochemetry(1999)), and Hagan et. al. (Biophys. J. 98 (2010), Adv. Chem. Phys. (2013), Rev. Phys. Chem. 66 (2015)) to simulate viral assembly. More recently, it was applied to study HIV-1 capsid assembly by Liu, et. al. (Bull Math Biol (2019).

We thank the reviewer for sharing these references with us. We have added these references to the Discussion section, paragraph 5 and hope that we have a more comprehensive representation of the literature on these two methodologies. The following text was added: “There are currently at least two methods developed to predict the mechanism of viral capsid assembly and packaging, namely the differential equation model and the coarse-grained modeling approach. The differential equation method was developed by Zlotnick et. al. [19,20] and Hagan et. al. [21-23]. More recently, both methods have been used to model HIV-1 Gag during viral assembly. Using the differential equation method, Liu et. al [24] predicted a nucleating event containing a 7-mer and then a 13-mer (7 and 13 connected hexagons) initiating formation of the Gag sphere [24]. Similarly, using the coarse-grained method, Tomasini et. al. [6] predicted capsid assembly dynamics allowing hexameric configurations for the Gag shell. By contrast, our model predicts a new assembly method where the curve of the sphere emerges as new Gag trimers are added, starting from a single initial trimer. From the apex of that single trimer, our model predicts addition of trimers following the 6-lune structure, to create the growing curvature of the forming sphere. As the curvature of the HIV-1 particle is forming, new Gag trimers would attach to the forming shell in locations allowed by the existing steric hindrance of the structure. As the sphere volume closes onto itself following the 6-lune pattern, the complete HIV-1 particle is formed (Fig. 3). In agreement with our model, Briggs et. al. also observed that as curved Gag hexameric lattices grow, hexamers quickly become tightly packed [25]. In the model proposed by Briggs et. al., regions of empty spaces (no Gag proteins present), significant variation in distances between adjacent hexagons and overlap of hexameric unit cells were needed to generate curved surfaces. These and other aspects of HIV-1 assembly and maturation have been reviewed earlier [26]”.

COMMENT #4.

“a 3-dimensional (3D) object constructed solely of regular hexagons cannot exist.” The meaning of this sentence may be not right. I can form a 3D capsid with hexagons by adjusting the gaps between hexagons. These irregular gaps may accommodate the curvature necessary to form an approximately spherical protein shell (E.O. Freed , Springer (2013), E.O. Freed Nat. Rev. Microbiol. (2015) )

We agree with the reviewer in that this sentence is not completely accurate. A cylinder, as shown in Fig. 1B, is in fact a 3D object constructed solely of regular hexagons. We have modified the sentence which now reads: “As a consequence, a polyhedron constructed solely of regular hexagons cannot exist”.

It is true that one could build a 3D curved object using hexagons. These hexagons however need to be irregular hexagons, as suggested by the reviewer, by adjusting “gaps” or the distance between adjacent hexagons. Even if we allow for the adjustment of distances between trimers, the collapsing effect shown in Supp. Fig. 2 means that after just 2 irregular hexagons, either a space or a pentagon needs to be placed. We added 2 sentences at the end of the first paragraph under Results stating: “The distance between adjacent trimers decreases from 5.21 nm down to 5.02, 4.43, 3.49 nm (Supp. Fig. 2). The mathematical proof to demonstrate how it is impossible to construct a sphere with regular hexagons is shown in Supp. Fig. 3”.

We have added a reference by E.O. Freed Nat. Rev. Microbiol (2015) at the end of paragraph 5 in Discussion.

COMMENT #5.

The sphere size and its relation to the radius of a final lipid Bilayer: y = 0.198x – 4.728. To my understanding, if x=10, the sphere size y<0. Therefore, this formula may be improved or authors may add some explanations about it.

There are still two similar formulas needed to be improved in this paper.

We agree with the reviewer and have included clarification statements for the x-axis range used to generate these 3 trend lines. 

In the case of the linear equation y = 0.198x – 4.728, we have added the following 2 sentences at the end of page 2: “The x-axis values used for Ico Sphere sizes were between 100 and 250 (arbitrary units) in Blender. The linear curve shown above might change outside of this x-axis range”.

For the polynomial equations in Fig. 6, we have added 2 sentences in the figure legend stating the x-axis range in which the equations apply: Panel A “The best-fit curve applies for diameters (x values) between 100 and 200 nm.” Panel B: “The best-fit curve applies for diameters (x values) between 134 and 135 nm.”

COMMENT #6.

“generating a 3D model of the HIV-1 MA shell in its spherical conformation using hexameric rings was impossible (Fig. 1C and D).” To my understanding, this method does not generate a spherical conformation using hexamers. But I believe that a spherical conformation can be generated by hexamers by adjusting gaps between hexamers.

As noted in the second paragraph of our “Discussion”, we address that hexagon-based spheres are possible as long as pentagons are introduced at regular intervals, with an example being a fullerene. However, regular hexagons (equal sides) cannot be fitted into a sphere, as demonstrated by Euler’s theorem. 

We then demonstrated the mathematical explanations showing precisely the geometrical limitations of a hexagon-based sphere. Our new Supp. Fig. 2 will help the reader understand how regular (equal sides) hexagons cannot be fitted into a sphere. A single irregular hexagon is possible (with a short side at 5.05 nm). However, extending an adjacent irregular hexagon from the shorter side will result in shorter distances between adjacent trimers leading to trimer volume overlap, thus collapse of the structure. We have also added a mathematical proof of this principle in the new Supp. Fig. 3. With regards to “adjusting gaps between hexamers”, our 6-lune hosohedra model is effectively the outcome of allowing gaps between trimers as we populate the surface of the sphere with trimers. The idea that the trimers have to exist in hexameric configuration is incompatible with the geometry of spheres. 

COMMENT #7.

Please check the reference “14. Cole IR (2015) Modelling CPV. 1-260.”

This reference has been removed from the manuscript.

---

## [Decision Letter · Decision Letter 1]

28 Oct 2019

Mathematical Determination of the HIV-1 Matrix Shell Structure and its Impact on the Biology of HIV-1

PONE-D-19-19426R1

Dear Dr. Marcet-Palacios,

We are pleased to inform you that your manuscript has been judged scientifically suitable for publication and will be formally accepted for publication once it complies with all outstanding technical requirements.

With kind regards,

Jamil S Saad, Ph.D

Academic Editor

PLOS ONE

Additional Editor Comments (optional):

Reviewers' comments:

Reviewer's Responses to Questions

**Comments to the Author**

1. If the authors have adequately addressed your comments raised in a previous round of review and you feel that this manuscript is now acceptable for publication, you may indicate that here to bypass the “Comments to the Author” section, enter your conflict of interest statement in the “Confidential to Editor” section, and submit your "Accept" recommendation.

Reviewer #1: All comments have been addressed

2. Is the manuscript technically sound, and do the data support the conclusions?

Reviewer #1: Yes

3. Has the statistical analysis been performed appropriately and rigorously? 

Reviewer #1: Yes

4. Have the authors made all data underlying the findings in their manuscript fully available?

Reviewer #1: Yes

5. Is the manuscript presented in an intelligible fashion and written in standard English?

Reviewer #1: Yes

6. Review Comments to the Author

Reviewer #1: Page 6, Last paragraph, 1st sentence "Our model suffers from several limitations" could be revised as follows. "It should be pointed out that our model is subject to some limitations."

7. PLOS authors have the option to publish the peer review history of their article (what does this mean?). If published, this will include your full peer review and any attached files.

Reviewer #1: No

---

## [Editor Report · Acceptance letter]

5 Nov 2019

PONE-D-19-19426R1 

Mathematical determination of the HIV-1 matrix shell structure and its impact on the biology of HIV-1

Dear Dr. Marcet-Palacios:

I am pleased to inform you that your manuscript has been deemed suitable for publication in PLOS ONE. Congratulations! Your manuscript is now with our production department. 

With kind regards,

on behalf of

Dr. Jamil S Saad 

Academic Editor

PLOS ONE